# Exhaled nitric oxide detection for diagnosis of COVID-19 in critically ill patients

**Matthew C. Exline**[1], **Milutin Stanacevic**[2], **Andrew S. Bowman**[3], **Pelagia-Irene Gouma**[4,5]*

**1** Department of Internal Medicine, Division Pulmonary, Critical Care and Sleep Medicine, The Ohio State University Wexner Medical Center, Columbus, Ohio, United States of America, **2** Department of Electrical & Computer Engineering, Stony Brook University, Stony Brook, New York, The United States of America, **3** Department of Veterinary Preventive Medicine, The Ohio State University, Columbus, Ohio, United States of America, **4** Department of Materials Science & Engineering, The Ohio State University, Columbus, Ohio, United States of America, **5** Department of Mechanical and Aerospace Engineering, The Ohio State University, Columbus, Ohio, United States of America

* gouma.2@osu.edu

**Data Availability Statement:** All relevant data are within the paper and its files.

**Funding:** NSF CBET #2029847 to PG. Orton Chair Endowment to PG.

## Abstract

### Background

COVID-19 may present with a variety of clinical syndromes, however, the upper airway and the lower respiratory tract are the principle sites of infection. Previous work on respiratory viral infections demonstrated that airway inflammation results in the release of volatile organic compounds as well as nitric oxide. The detection of these gases from patients' exhaled breath offers a novel potential diagnostic target for COVID-19 that would offer real-time screening of patients for COVID-19 infection.

### Methods and findings

We present here a breath tester utilizing a catalytically active material, which allows for the temporal manifestation of the gaseous biomarkers' interactions with the sensor, thus giving a *distinct breath print* of the disease. A total of 46 Intensive Care Unit (ICU) patients on mechanical ventilation participated in the study, 23 with active COVID-19 respiratory infection and 23 non-COVID-19 controls. Exhaled breath bags were collected on ICU days 1, 3, 7, and 10 or until liberation from mechanical ventilation. The breathalyzer detected high exhaled nitric oxide (NO) concentration with a distinctive pattern for patients with active COVID-19 pneumonia. The COVID-19 "breath print" has the pattern of the small Greek letter omega (). The "breath print" identified patients with COVID-19 pneumonia with 88% accuracy upon their admission to the ICU. Furthermore, the sensitivity index of the breath print (which scales with the concentration of the key biomarker ammonia) appears to correlate with duration of COVID-19 infection.

### Conclusions

The implication of this breath tester technology for the rapid screening for COVID-19 and potentially detection of other infectious diseases in the future.

**Competing interests:** The authors declare there is a provisional patent on the discussed technology.

## Introduction

A common feature of respiratory viral infections is the release of inflammatory cytokines. These cytokines led to the production and release of volatile organic compounds (VOC), nitric oxide (NO), and ammonia ($NH_4$). Our previous work demonstrated that use of nanosensor systems are able to detect these exhalents and have the potential for early detection and disease monitoring of patients with viral respiratory infections [1]. However, the specific signature of a specific patient population with a given virus still needs to be developed.

The World Health Organization (WHO) declared COVID-19 disease in March 2020 [2]. The coronaviruses known to infect humans generally only caused mild upper respiratory tract infectious symptoms. They are also known to delay the innate immune response to infection, and they have affinity for primary epithelial cells [3]. SARS-CoV-2 belongs to the distinct group of coronaviruses known as beta CoV [3]; COVID-19 is the clinical syndrome that develops as a result of the first pandemic caused by a coronavirus [4]. COVID-19 is manifested in variable symptoms: mild upper respiratory symptoms such as cough, sore throat, anosmia, and myalgia; moderate symptoms of dyspnea; and severe symptoms including hypoxemia and respiratory failure [5]. The in severe cases the disease often progresses over the course of 5–12 days and symptoms especially ongoing organ failure may persist after the infection as clear.

The gold standard for diagnosis of COVID-19 are FDA-approved molecular tests, but these suffer from low detection accuracy for early COVID-19 infection and persistent positive results after infection has resolved [6]. In this work, we present a novel breathalyzer technology that utilizes a single selective, resistive chemosensor made of a catalytically active, semiconducting material, targeting NO and ammonia molecules in breath. The use of a single sensor allows for rapid analysis of data and diagnostic results. This then allows the device/tester to detect the distinct signature (breath print) of COVID-19, non-invasively, in exhaled breath, within 15 seconds. The technology presented here is distinct from other artificial olfactory systems. Traditional "electronic nose" technology utilizes non-selective sensors in an array that swells and shrink to change the electrical properties of a transducer based on exposure to VOCs and then utilize advanced machine learning and feature extraction algorithms to separate the signal in an attempt to detect COVID-19 [7]. This technology, in contrast, relies on materials science-controlling polymorphism and phase tailoring using nanotechnology- to detect and measure the specific targeted biomarkers in the complex breath environment. This study focused on use of *a novel and viable pathway* towards the early and rapid detection of infection COVID-19 using this nanosensor system.

## Methods

### Breath detector device

The COVID-19 breathalyzer is an electronic device, which uses a single *catalytically active, resistive sensor* that is highly selective to NO. The sensitivity of the $\gamma$-phase tungsten trioxide ($WO_3$) sensor to NO, selectivity and response in the presence of various interfering compounds have been demonstrated before [8] and are shown here for the specific conditions of this study, simulating human exhaled breath having various concentrations of NO and of the most abundant VOCs in breath: acetone, isoprene, and ammonia (**Fig 1A**).

The catalytic /sensing film was produced by means of sol-gel processing using Tungsten Alkoxide precursors (Fisher Scientific). Following aging for 2 days, the gels were calcined at 550˚C for four hours producing nanoscale powders of pure-$WO_3$. The crystal structure of the materials was determined using X-ray powder diffractometry (XRD Bruker D8) (not shown here). Then, a solution was made of 0.01 grams of the calcined powders in ethanol and

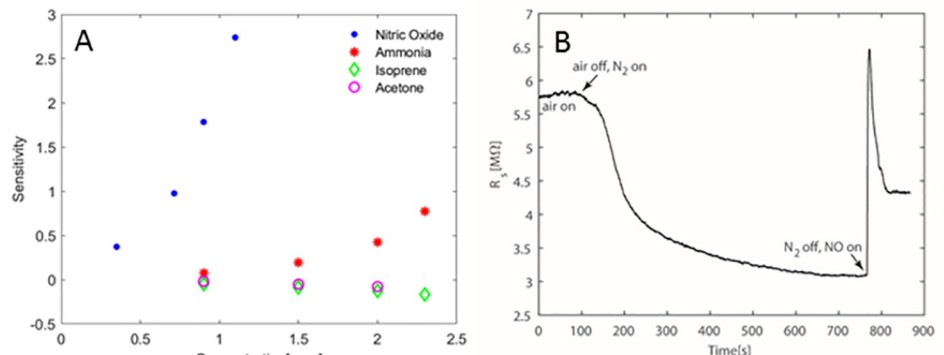

**Fig 1. Breathalyzer sensitivity for NO. A.** To demonstrate sensitivity to NO, known concentrations of NO, ammonia, isoprene, and acetone were put into breath bag and "exhaled" over circuit. The circuit showed excellent sensitivity to NO and minimal interaction with ammonia at higher concentrations and no signal from acetone and isoprene. **B.** Specificity of circuit for NO versus $N_2$ demonstrates that circuit does not interact with $N_2$, but has robust detection of NO.

1microliter of the solution was deposited on alumina substrates (of 3mmx3mm dimensions) on which platinum interdigitated electrodes were printed, until a meniscus bubble was formed. This step was followed until the desired film thickness was achieved. Electrical connections between the sensor substrate and the TO-8 substrate were made using gold wires. For the sensor calibration, a custom gas flow bench was used, consisting of MKS 1179a mass flow controllers. For simulating the breath gas, OSHA grade D breathing air gas cylinders and other specialty gases (acetone; isoprene; ammonia; NO) in custom (ppm-ppb) compositions were purchased by Praxair.

For integration of the sensor into a breathalyzer, there is a circuit for the conversion of the resistance of the sensor that is proportional to the NO concentration to a digital value [7, 8]. In the design, there is a discrete implementation of the front-end readout circuit that achieves satisfactory precision for a wide range of measured resistances. The output voltages of the front-end readout electronics are interfaced to a National Instruments data acquisition card (NI-DAQ), NI 6259, and converted to digital domain and displayed on the screen of a PC. When a breath is introduced into the breathalyzer, the sensor interacts with the gas molecules producing an electrical output/signal.

## Clinical study protocol

The Ohio State University Biomedical Sciences Institutional Research Board (IRB) and Institutional Biosafety Committee (IBC) approved study protocol according to local COVID-19 research protocols. Patients admitted to the intensive care unit (ICU) requiring mechanical ventilation participated in this study. Inclusion criteria included acute respiratory failure requiring endotracheal intubation and mechanical ventilation. Exclusion criteria included mechanical ventilation for > 72 hours at time of enrollment, inability to obtain informed consent, or prisoner status. All patients underwent a molecular (PCR based) COVID-19 test on admission to hospital per policy. Study personnel screened patients under a partial HIPAA waiver. Due to pandemic research protocol and minimal risk associated with this study, verbal or written consent were acceptable for either patient or legally authorized representative. In the case of remote (verbal) consent, an IRB approved telephone script for consent process was utilized and consent was documented by study personal. After informed consent of either patient or their surrogate decision maker, exhaled air samples were collected on study days 1,

3, 7, and 10 approximately between the hours of 8am and 2pm or until subject was liberated from mechanical ventilation. Clinical data collected included basic demographic information, reason for admission to ICU and presence of high-risk medical conditions for COVID-19, and basic physiologic data to determine the patient's Sequential Organ Failure Assessment Score (SOFA Score) on each study day [9, 10].

Samples were collected from the exhalation port of the ventilator in 1-liter breath bags (Tedlar bags, CEL Scientific). A HEPA filter (Teleflex Hudson RCI Gibeck Iso-Gard HEPA Light) was placed over the exhalation port per institutional protocol to minimize potential aerosolization of virus. All studies personnel wore appropriate personal protective equipment, including respiratory, as indicated by hospital infection prevention guidelines. After collection, samples were transported to BSL3 level lab for analysis. Samples were collected between 8am and 2pm. All testing was done within 4 hours of sample collection (**Fig 2**).

## Statistical analysis

A total sample size of 46 patients (23 COVID-19 positive, 23 COVID-19 negative control patients) were recruited based a priori assumptions to give an 80% power to detect a 50% increase in exhaled NO in COVID-19 infected individuals with an $\alpha = 0.05$. Discrete variables were analyzed using Pearson Chi-square test. Continuous variables using either Student's t-test or Wilcoxon Rank Sum analysis depending on distribution. All analysis performed on JMP Pro 14.0.0 (SAS Institute Inc.).

## Results

### Breath detector device validation

Using the breathalyzer system, we plotted the signature pattern of all mechanically ventilated patients. Breath is a complex gas environment and NO and ammonia appear to be present in significant amounts in a variety of patients. We identified several stereotypical patterns, which we termed the NO-pattern, $NH_3/O_2$-pattern, and the Omega-pattern. Specific to COVID-19 infections, a distinct breath print that appears to give three peaks on the detector, like the small Greek letter omega () was found (**Fig 3**). The omega pattern results from the interaction between oxygen, NO, and ammonia. In the oxidation of ammonia, "the reduction of the product of extensive oxidation by the initial oxidisable substance" can take place as follows: $4NH_3 + 6NO = 5N_2 + 6H_2O$ [11]. This reaction is responsible for the first reducing step observed as soon as the exhaled breath reaches the sensor, which lowers the baseline resistance of the sensor which reaches a minimum (S1). Next there is an oxidative step (S2 peak) corresponding to *ammonia oxidation to NO*. The second reducing and final step (S3) is again due to the reaction of any remaining ammonia to newly formed NO, and the sensor recovers in air reaching its baseline. It is interesting to note that these redox processes on the sensor are not favored at temperatures below 300˚C, while the breathalyzer probe operates at temperature of 300˚C. To verify that the omega pattern was due to this combination of reduction and oxidation of ammonia and NO in exhaled breath, we reproduced the pattern *in-vitro*. Introducing $N_2$ to lower the resistance from the baseline value followed by adding NO to increase it and replacing it with $N_2$ before exposing the sensor to air to reproduce the omega pattern (**Fig 1B**).

This distinct profile reflects the ability of the material used (pure $WO_3$) to act both as sensor and as a catalyst for redox reactions involving ammonia and NO in exhaled breath. While the sensor is selective to NO, the interaction of ammonia gas with the nanostructured pure $WO_3$ sensing element results in the non-selective catalytic oxidation of ammonia to NO, which accounts for the distinct peak (S2) in the pattern obtained. Other workers using Diffuse Reflectance Fourier Transform Spectroscopy (DRIFTS) [12] studied ammonia gas adsorption on

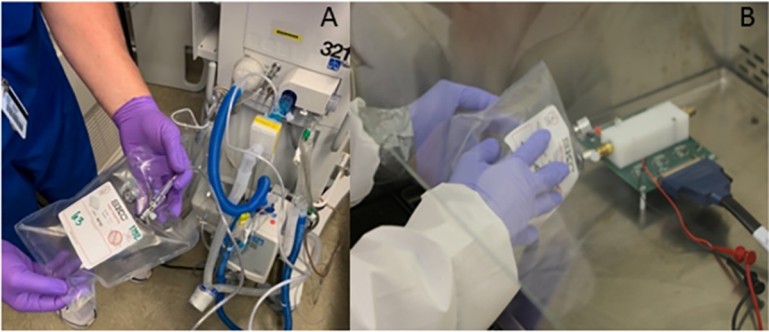

**Fig 2. Breath bag collection technique. A.** Breath bag was connected to exhalation port of ventilator at approximately 1 liter of exhaled gas was collected over 3–5 respiratory cycles. **B.** Samples were brought to a BSL 3 lab and attached to breathalyzer circuit in hood.

pure WO3. It was concluded that oxidation of gaseous or adsorbed ammonia species to NO occurred on the surface of $WO_3$ [12]. Thus, the pure -$WO_3$ sensor in the breathalyzer catalyzes the oxidation of $NH_3$ to NO ($NH_3$ -> NO + $H_2O$) producing an "oxidative" response which is manifested as a peak (increase) in the NO concentration.

If this was the only interaction between breath gaseous biomarkers and the sensor, then we would simply need to add a selective ammonia sensor (like the one we developed previously [13]) to this NO sensor and the two-sensor array would be sufficient to provide the relative ratio of the two biomarkers that would signal the disease. However, it is not feasible to detect COVID-19 by two selective sensors. What is unique to this breathalyzer technology and makes for a rapid and reliable COVID-19 test is its ability to map the catalytic reactions while occurring in the breath gas mixture, on the sensor, at the specific operating conditions (set temperature) of the sensor. What should also be noted is that it is unlikely for the omega breath pattern to be attributed to the oxidation of other organic compounds to CO, ketones, alcohols, etc. [14] as the sensor response to these gases was found to be flat, meaning there was no sensitivity to these at all [8].

## Clinical validation

A total of 46 patients agreed to participate in the study. After sample collection, seven patients' data was excluded due to us failing to detect the end-of-the-life of the sensor on time, leaving 39 patients for analysis. **Table 1** contains their demographic information and past medical history. Overall, patients were similar between COVID-19 and other critically ill control patients with only a higher rate of diabetes in the COVID-19 positive patients that was statistically significant and a non-significant higher rate of obesity. Both of these findings are likely related to obesity and diabetes being strong risk factors for severe COVID-19 infection (**Table 1**).

Demographic information on 46 patient validation cohort included. All values presented as either median and IQ range or number (%) as appropriate. Past medical history of heart disease, lung disease, hypertension, obesity, diabetes, active cancer, cirrhosis, and end-stage renal disease based on clinician review of presenting history and physical. SOFA scores calculated on day on study day 1. Wilcoxian testing showed only significant difference between in prevalence of diabetes between COVID-19 and control patients. Survival was defined as survival to hospital discharge.

Analysis with our breath detector showed that patients with early COVID-19 infection (within 72-hours of onset of respiratory failure) demonstrated the stereotypical omega pattern (**Fig 3**). The use of COVID-19 breath analysis was helpful in rapidly screening for patients

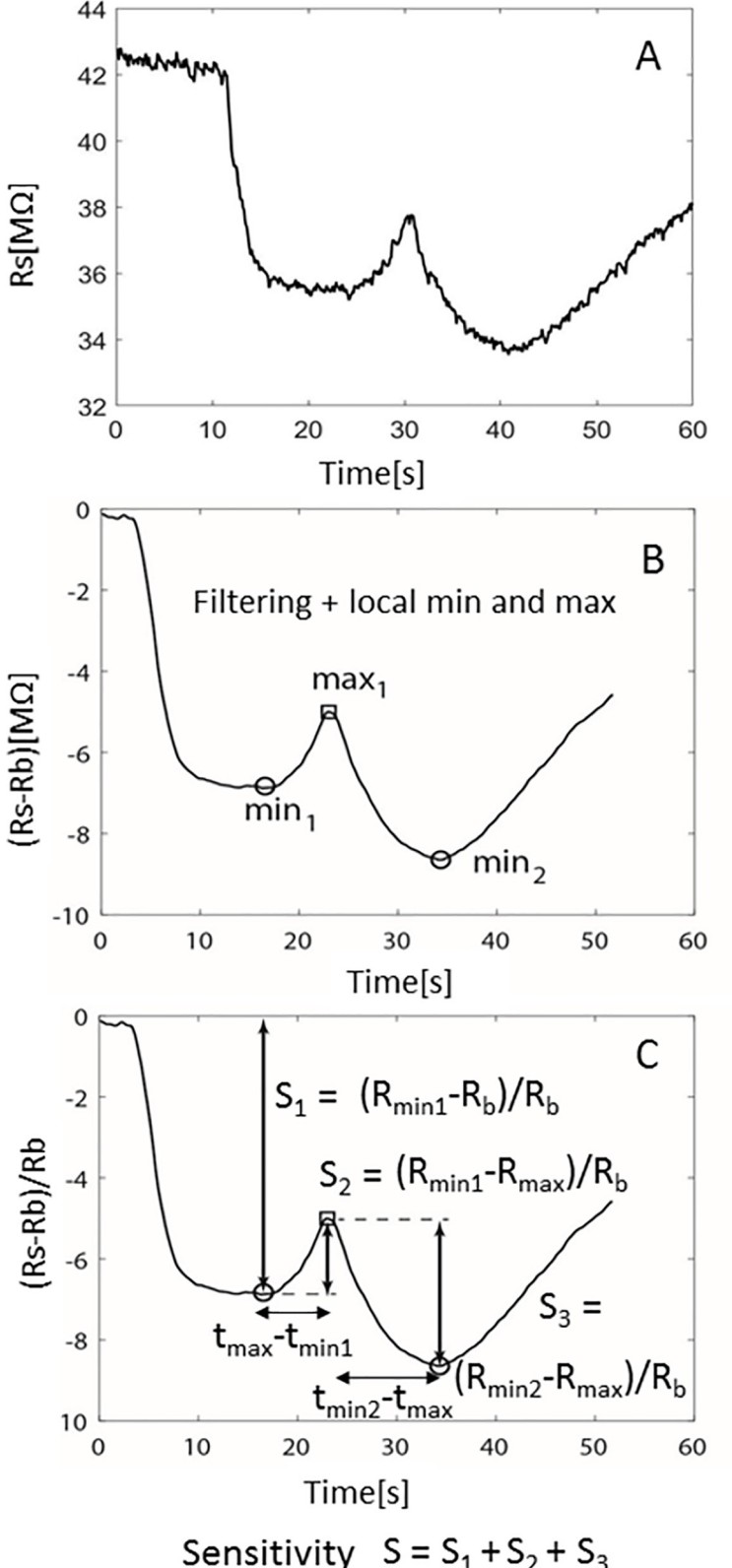

**Fig 3. Breathalyzer sensitivity for NO. A.** Omega Pattern seen in critically ill patients with severe COVID-19 pneumonia. We start in air and have a baseline. **B.** As soon as breath reaches the sensor and until the ammonia is

converted to NO there is a first minimum formed (min1) and after the maximum (for NO) and until the sensor is exposed to the air again, there is a second minimum (min2). **C.** Sensitivity are determined from baseline to min1 (S1), min1 to intermediate peak (S2), and min2 to intermediate peak (S3). S2 is used as marker of COVID-19 severity.

with severe respiratory failure and excluding those without COVID-19 infection on mechanical ventilation. Overall, of 16 patients positive for COVID-19 on study day 1, 14 (88%) demonstrated the omega pattern (p<0.0001) (**Table 2**). The negative predictive value of the breathalyzer was excellent at 90%. (**Table 3**). The two patients with false negative breathalyzer results were both relative remote from onset of COVID symptoms (8 and 20 days respectively) when they had respiratory compromise and both were diagnosed with bacterial pneumonia suggesting that their respiratory failure might not have been due to virus. The four patients with false positive breathalyzer tests were admitted for a variety of diagnosis (stroke, pneumonia, and cirrhosis). That two of the patients had cirrhosis suggests that the elevated ammonia from their underlying cirrhosis may have been responsible for false positive result. Obviously, it is impossible to draw any conclusions on any discordant results given the sample size.

Overall, there was a trend for resolution of the omega pattern as patients' clinical infection resolved. The amplitude of the omega pattern (S2 on **Fig 3C**) was associated with days from clinical onset of COVID-19 pneumonia among all COVID-19 patients ($R^2$ = 0.12, p = 0.037) (**Fig 4A**). Generally, patients would transition from the omega pattern to a single peaked pattern, the NO pattern, through the course of their critical illness. Patients with persistent need for the ventilator had a trend to continuing to display the omega pattern likely representing ongoing inflammation and lung injury. Among the COVID-19 positive patients that remained on the ventilator for the 10-days of the study, there was a general trend in the decrease of the amplitude of the omega S2 peak over the course of their illness until transitioning to the NO pattern. (**Fig 4B**).

## Discussion

This is the first work to our knowledge to demonstrate use of a nanosensor breathalyzer system to detect a viral infection from exhaled "breathe prints". The test is non-invasive and rapid in determining the result. Due to epidemiologic concerns and regulations regarding exposure of

**Table 1. Patient characteristics of ICU patient population.**

|  | COVID Positive (n = 23) | Control (n = 23) |
|---|---|---|
| Age (median IQ) | 61 (58, 74) | 65 (54, 72) |
| Male Sex | 14/23 (61%) | 11/23 (48%) |
| Heart Disease | 9/23 (39%) | 13/23 (56%) |
| Lung Disease | 7/23 (30%) | 13/23 (56%) |
| Hypertension | 17/23 (74%) | 21/23 (91%) |
| Obesity (BMI > 30) | 13/23 (56%) | 8/23 (35%) |
| Diabetes* | 12/23 (52%) | 5/23 (22%) |
| Active Cancer | 3/23 (13%) | 8/23 (35%) |
| Cirrhosis | 1/23 (4%) | 4/23 (17%) |
| End-Stage Renal Disease | 6/23 (26%) | 3/23 (13%) |
| SOFA (median IQ) | 9 (8, 11) | 10 (7, 13) |
| Survival | 17/23 (73%) | 12/23 (52%) |

**Table 2. Omega pattern clinical performance study day 1.**

| Pattern | COVID-19 Group | Control Group |
|---|---|---|
| Omega | 14 | 4 |
| Non-Omega | 2 | 19 |

research staff, for purposes of the research trial the analysis was not done at the bedside; but in clinical practice, this could be done easily.

The concept of this COVID-19 breath test evolved from the advanced sensor technologies for breath diagnostics based on the crystallo-chemical principle of selective gas detection that were developed by our group and have existed for almost two decades now [15–20]. Most well-known include single selective sensors for each specific biomarker, e.g. NO for asthma [21], or acetone for metabolic disorders [17]. Infectious diseases are typically characterized by more than one biomarker. Influenza is marked by the release of pro-inflammatory cytokines, which results in the generation of a number of volatile products that infiltrate the lungs. These products include a number of Volatile Organic Compounds (VOCs)-predominantly Isoprene and Nitric Oxide (NO) as demonstrated by Dweik and colleagues [22]. Previous work by our group demonstrated the concept of the flu breathalyzer, a portable, handheld sensor system which targets the biomarkers for the infection from the influenza virus specifically isoprene and Nitric Oxide implement one selective sensor for each biomarker [1].

Unlike the electronic nose technologies which utilize arrays of non-selective sensors and sample the whole breath for patterns drawn by machine learning algorithms [23], this selective chemosensing technology provides an electronic signature, based on the presence of the particular chemical compound (biomarker) and its distinct concentration in a single breath exhaled. The breath print of COVID-19 identified in this work reflects the host response to the specific virus it is not affected by the external environment at which a measurement is made, as the sensor response is specific to the analyte of interest. Furthermore, we avoid the interaction of other VOC such as acetone, which may be influenced by the patient's underlying medical conditions. The sensor employed in this COVID-19 breath test is unique and different than conventional resistive sensors as it captures, in temporal information, the interaction and relative ratio of the two distinct gases (NO and Ammonia) that cannot be captured by two selective sensors. It also differs from gas chromatography (GC) as ammonia is often absorbed by the stainless steel used for GC detection [24]. This one-of-a kind technology is based on the semiconducting, catalytic and gas sensing characteristics of pure $WO_3$ and the specific redox reactions occurring between the two biomarkers in the presence of the sensor.

This technology enables the potential rapid detection of the biomarkers that manifest COVID-19 disease in a single step. Given the concerns the current tests that are employed for COVID-19 detection, "neither PCR nor immunoassay techniques are ideal" due to the time needed to process and difficulty in large scan rollouts [25]. PCR tests are accurate if enough virus load exists in the area that is being swabbed but they are still slow, often taking hours to run, and laborious. While PCR based detection platforms are the gold standard of COVID-19

**Table 3. Omega pattern diagnostic performance study day 1.**

| Sensitivity | 88% |
|---|---|
| Specificity | 83% |
| Positive Predictive Value | 78% |
| Negative Predictive Value | 90% |
| Accuracy | 85% |

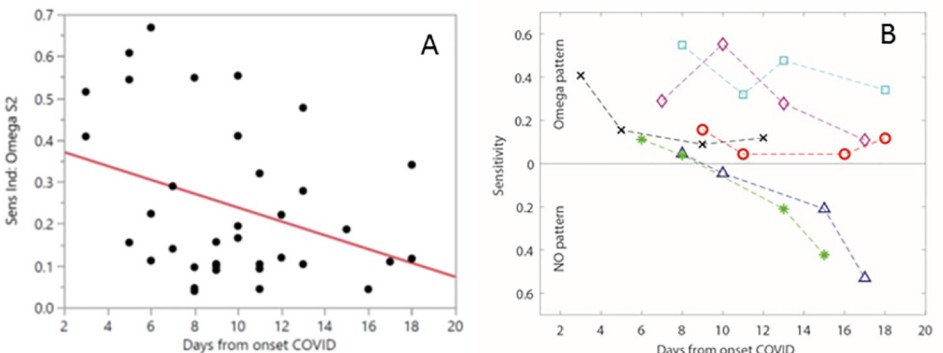

**Fig 4. Amplitude of omega pattern over COVID course. A**. The S2 peak as described previously measured amplitude of omega pattern. Plotting amplitude of S2 by days from COVID-19 onset defined as either day of first symptoms if patient able to provide or as days from first positive test if unable to determine symptom onset ($R^2 = 0.12$, p = 0.04). **B**. For select patients requiring mechanical ventilation over 10 days, amplitude of omega is plotted over days from COVID onset.

testing, their sensitivity in clinical practice can vary between 71–98% [26] comparable with our reported sensitivity of 88%.

There are several limitations of our current study. We were evaluating patients that were critically ill and likely represented the most severe cases of COVID-19 pneumonia. All patients were tested for SARS Co-2 virus, but we did not screen for other coronavirus or respiratory viruses. It is possible, that other coronaviruses will have a similar signal. Due to safety concerns around aerosol generating procedures, we did not test patients after they were extubated so we were unable to document return to baseline in all patients, but did see in a number of patients as they recovered from COVID-19. We also had to exclude the data from seven patients. Our sensor has a limited lifetime as the baseline of the sensor drifts due to the contamination of the sensor surface and the sensor stops responding to NO. The breath was collected for seven patients with the sensor that reached the end of the life and was unknown due to blinding of the research personal. After this incident, end-of-sensor-life detection has been incorporated in breathalyzer software.

Despite these limitations, the use of breathalyzer technology to rapidly diagnose patients with respiratory infections has the potential to greatly improve our ability to rapidly screen both patients and asymptomatic individuals. This study is the first to our knowledge to show the practical application of this emerging technology in a homogenous group of patients with a single infection. Future studies are needed to determine what other disease or infections could benefit from this technology.

## Supporting information

**S1 File.**
(PDF)

## Acknowledgments

The authors wish to acknowledge the hard work of Emily Robart, Sarah Karow, Brent Oleksak, and Adam Soliman for specimen and data collection. The authors wish to acknowledge Dillon McBride, Sarah Lauterback, Yasha Karimi, Fateh Mikaeili, Milind Pawar, and Owen Abe for sample analysis and device development.

## Author Contributions

**Conceptualization:** Milutin Stanacevic, Andrew S. Bowman, Pelagia-Irene Gouma.

**Data curation:** Matthew C. Exline, Milutin Stanacevic, Andrew S. Bowman, Pelagia-Irene Gouma.

**Formal analysis:** Matthew C. Exline, Milutin Stanacevic, Andrew S. Bowman, Pelagia-Irene Gouma.

**Funding acquisition:** Pelagia-Irene Gouma.

**Investigation:** Matthew C. Exline, Milutin Stanacevic, Andrew S. Bowman, Pelagia-Irene Gouma.

**Methodology:** Matthew C. Exline, Milutin Stanacevic, Andrew S. Bowman, Pelagia-Irene Gouma.

**Project administration:** Matthew C. Exline, Pelagia-Irene Gouma.

**Resources:** Pelagia-Irene Gouma.

**Software:** Milutin Stanacevic.

**Supervision:** Matthew C. Exline, Milutin Stanacevic, Andrew S. Bowman, Pelagia-Irene Gouma.

**Validation:** Matthew C. Exline, Milutin Stanacevic, Andrew S. Bowman, Pelagia-Irene Gouma.

**Visualization:** Milutin Stanacevic.

**Writing – original draft:** Matthew C. Exline, Milutin Stanacevic, Andrew S. Bowman, Pelagia-Irene Gouma.

**Writing – review & editing:** Matthew C. Exline, Milutin Stanacevic, Andrew S. Bowman, Pelagia-Irene Gouma.

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
