## [Decision Letter · Decision Letter 0]

21 Jul 2021

PONE-D-21-20035

A Breath Tester for COVID-19

PLOS ONE

Dear Dr. Exline,

Thank you for submitting your manuscript to PLOS ONE. After careful consideration, we feel that it has merit but does not fully meet PLOS ONE’s publication criteria as it currently stands. Therefore, we invite you to submit a revised version of the manuscript that addresses the points raised during the review process.

We look forward to receiving your revised manuscript.

Kind regards,

Aleksandar R. Zivkovic

Academic Editor

PLOS ONE

2. Thank you for including your ethics statement: "Institutional Research Board (IRB) and Institutional Biosafety Committee (IBC) approved study protocol according to local COVID-19 research protocols. Patients admitted to the intensive care unit (ICU) requiring mechanical ventilation participated in this study. Inclusion criteria included acute respiratory failure requiring endotracheal intubation and mechanical ventilation. Exclusion criteria included mechanical ventilation for > 72 hours, inability to obtain informed consent, or prisoner status. " 

b) Please provide additional details regarding participant consent. In the ethics statement in the Methods and online submission information, please ensure that you have specified what type you obtained (for instance, written or verbal, and if verbal, how it was documented and witnessed). If your study included minors, state whether you obtained consent from parents or guardians. If the need for consent was waived by the ethics committee, please include this information.

3. Please modify the title to ensure that it is meeting PLOS’ guidelines (https://journals.plos.org/plosone/s/submission-guidelines#loc-title). In particular, the title should be "specific, descriptive, concise, and comprehensible to readers outside the field" and in this case we feel it is not informative and specific about your study's scope and methodology.

4. We note that you have a patent relating to material pertinent to this article. Please provide an amended statement of Competing Interests to declare this patent (with details including name and number), along with any other relevant declarations relating to employment, consultancy, patents, products in development or modified products etc. Please confirm that this does not alter your adherence to all PLOS ONE policies on sharing data and materials, as detailed online in our guide for authors http://journals.plos.org/plosone/s/competing-interests by including the following statement: "This does not alter our adherence to  PLOS ONE policies on sharing data and materials.” If there are restrictions on sharing of data and/or materials, please state these. Please note that we cannot proceed with consideration of your article until this information has been declared.

7. Please upload a copy of Figure 5, to which you refer in your text on page 17. If the figure is no longer to be included as part of the submission please remove all reference to it within the text

Reviewers' comments:

Reviewer #1: Exline at al report on the application of a breath tester for the diagnosis of COVID19 in ICU patients. They found a specific breath print pattern (i.e. omega pattern) to predict COVID19 with 82% accuracy and they conclude that this technology could help for the rapid screening of COVID19.

On my opinion the main limitation of the study is that it is unclear to what extent the omega pattern really predicts SARS-CoV2 infection rather than lung or systemic inflammation, that is also known to have a significant impact on exhaled breath composition. In other words, the observed differences may be attributable to differences in the inflammatory responses of enrolled patients, and not to SARS-CoV2 presence. In addition to that, patients have been poorly characterized in term of other clinical and biochemical parameter (Table 1), and their association with the omega pattern has not been evaluated. Lastly, I do not see much interest in applying this test in ICU patients. I would be more interested in applying it in asymptomatic subjects for their early screening. In any case, results should be replicated in subjects without severe COVID19 infection, versus asymptomatic patients without COVID19.

Reviewer #2: This is an interesting study in which a sensor was used to measure 'signatures' of NO and NH3 from the breath of COVID patients vs controls. I think this could eventually be published subject to some revisions.

1. The phrasing 'sensor array' is not accurate. An array implies multiple different sensors operating in parallel, not a single catalytica sensor.

2. An area under the curve plot should be shown.

3. The raw sensor data for all patients should be shown. It is not clear what happens in cases of false positives and negatives.

4. The explanation of the omega pattern is possible but data supporting the explanation is minimal. Does introducing NO and NH3 to the sensor surface result in an omega pattern.

5. The speculation about having two sensors selective for ammonia and NO not working as well is not clear. I'd expect the more sensors the better the results.

Reviewer #3: The work presents separation of patients with COVID-19 from patients with other diseases through measurements of nitric oxide (NO) and ammonia (NH3) in their breath. My comments are appended below.

1. Did you perform the tests at a certain time of the day, which might have affected the concentration of the volatiles that you detected? It would have been very informative if you could test daily variation of the volatiles (specifically the NO) and if it is different in patients compared to healthy individuals and/or other patient groups.

2. How do you interpret those patients with COVID-19 that have rather constant omega amplitude at different days from the onset of disease, like the one indicated with red circles in Figure 4B?

3. NO indication within "NO pattern" label at Figure 4B mixes up with nitric oxide, but it is not standing for that, as far as I understood. Also, does the "NO pattern" side of the figure in Figure 4B have negative sensitivity values? They are below zero, but the values are non-negative. Is it shown like that on purpose, or by mistake? If it is shown like that on purpose, it means that the sensitivity at the "omega pattern" side decreases with the progress of time and then it increases at the "NO pattern" side... Is it the case?

4. It is written on page 5, line 66 that “The technology presented here is distinct from other artificial olfactory systems..." Please cite the respective studies there.

5. Inclusion to the study required mechanical ventilation (page 6, line 104) and exclusion required mechanical ventilation more than 72 hours (page 6, line 106). So, did inclusion require mechanical ventilation up to 72 hours? What were the exact criteria? Did you include those that require mechanical ventilation only up to 3 days and collected sample on days 1, 3 (days during mechanical ventilation), and 7 and 10 (days after mechanical ventilation)? That issue seems to require clarification...

6. About the reaction equation written on page 8, lines 138-139: As far as I know, you cannot write non-integer numbers as the number of molecules entering a reaction or the number of molecules generated as products. In relation, does NH3 oxidize to NO, or reacts with NO as shown in lines 138-139? Both are mentioned in the text. Do both occur in this sensor? Besides, the reaction equation written on page 8, lines 138-139, is not present in the cited reference (reference number 10, Gouma et al. 2010 IEEE Sensors J 10:49) as far as I have seen.

7. It is written on page 8, lines 146-147 that "...these redox processes on the sensor are not favored at temperatures below 300°C." What was the measurement temperature?

8. It is written on page 8, lines 149-150 that WO3 "... act both as sensor and as a catalyst for redox reactions involving ammonia and NO in air." Do you mean NO in the air or in the exhaled breath there?

9. Figure 5 is cited at two places in the text. However, there is no Figure 5 or its caption, please add that. Also, can you comment on the observation mentioned on page 9, lines 172-173, i.e., high NO in breath samples of patients with other diseases than COVID-19?

10. It is written on page 9, lines 176-177 that "...seven patients’ data was excluded due to us failing to detect the end-of-the-life of the sensor on time..." Did you lose the collected data, or else?

11. Was the higher incidence rate of diabetes in the COVID group related to its being a risk factor?

12. It is written on page 11 lines 204-206 that "Most well known include single selective sensors for each specific biomarker, e.g. NO for asthma, or acetone for metabolic disorders." Please cite the relevant references there.

13. Please check reference 9, which is cited for the DRIFTS study on page 8, line 154, and for a clinical study, earlier on page 7, line 113. It is not involving DRIFTS study. Details of reference 9 that is given at the references section: Lambden S, Laterre PF, Levy MM, Francois B. The SOFA score-development, utility and challenges of accurate assessment in clinical trials. Crit Care. 2019;23(1). doi: ARTN 374 10.1186/s13054‐019‐2663‐7. PubMed PMID: WOS:000501779700001.

14. Please include a discussion including comparison of your sensor with the other breath tests that are developed for COVID-19, and for testing the same analytes.

15. About language and other issues:

Please check the language.

Breath bad is written at the caption of Figure 1. Is it correct, or did you mean breath bag?

Please indicate at axis labels of Figure 3 that time is in seconds.

Please correct Ieee as IEEE on page 15, line 293.

Also, please write the full-length form of abbreviations where they first appear in the text and at the caption of figures and tables, when abbreviations are present in tables and figures.

COVID needs to be replaced with COVID-19.

6. PLOS authors have the option to publish the peer review history of their article (what does this mean?). If published, this will include your full peer review and any attached files.

Reviewer #1: No

Reviewer #2: No

Reviewer #3: **Yes: **Yekbun Adiguzel

---

## [Author Response · Author response to Decision Letter 0]

4 Sep 2021

We thank the editors and reviewers for their thoughtful comments which we feel greatly improved the clarify of our manuscrip. Please refer to the extensive documentation in our response to reviewers section uploaded with submission.

---

## [Editor Report · Decision Letter 1]

7 Sep 2021

Exhaled nitric oxide detection for diagnosis of COVID-19 in critically ill patients

PONE-D-21-20035R1

Dear Dr. Exline,

We’re pleased to inform you that your manuscript has been judged scientifically suitable for publication and will be formally accepted for publication once it meets all outstanding technical requirements.

Kind regards,

Aleksandar R. Zivkovic

Academic Editor

PLOS ONE

---

## [Editor Report · Acceptance letter]

21 Oct 2021

PONE-D-21-20035R1 

Exhaled nitric oxide detection for diagnosis of COVID-19 in critically ill patients 

Dear Dr. Exline:

I'm pleased to inform you that your manuscript has been deemed suitable for publication in PLOS ONE. Congratulations! Your manuscript is now with our production department. 

Kind regards, 

on behalf of

Dr. Aleksandar R. Zivkovic 

Academic Editor

PLOS ONE